# Wood-Splitter-Related Upper-Limb Injuries: A Single-Centered Case-Series Study

**DOI:** 10.3390/ijerph191811507

**Published:** 2022-09-13

**Authors:** Arisa Aoyagi, Osamu Nomura, Norihiro Sasaki, Yuki Fujita, Nana Ichikawa, Yoshiya Ishizawa, Yasuyuki Ishibashi, Hiroyuki Hanada

**Affiliations:** 1Department of Emergency and Disaster Medicine, Hirosaki University, Hirosaki 036-8562, Japan; 2Department of Orthopaedic Surgery, Hirosaki University, Hirosaki 036-8562, Japan

**Keywords:** hand injuries, injury prevention, wood splitter

## Abstract

(1) Background: Injuries to the upper limbs during wood splitting can affect social and economic life. We aimed to describe the clinical information concerning these injuries in Japan. (2) Methods: We identified patients from our patient database from April 2015–November 2021 and extracted data from their medical records, which includes age, gender, occupation, month, time and location of the injury, diagnosis, duration of hospitalization, ICU admission, treatment interventions including surgery, outcome, and medical costs. (3) Result: Seventeen cases were identified. Most of the patients were male (*n* = 15), with median age being 68 years old. Regarding the patients’ backgrounds, six were apple farmers and three were unemployed. Injuries to the index finger was most common (*n* = 9), followed by injuries to the thumb in five cases (*n* = 5). Most of the incidents occurred at home or on the patient’s farm estate. No injuries were due to incidents at work. (4) Conclusion: The wood splitter-related injuries required long-term treatment and frequently damaged the thumb, a functionally important digit. All the injuries were sustained during non-occupational use of a wood splitter. Therefore, we suggest that safety training should be provided to prevent traumatic injuries when these products are being sold.

## 1. Introduction

People have shifted towards “stay-at-home” lifestyles in response to the COVID-19 pandemic and have implemented new home appliances to promote their well-being at home [1]. Wood-fired stoves have become popular heating equipment, as the stoves are suitable under the carbon-neutral policy for preventing global warming in the post-COVID-19 era [2]. However, it is currently difficult to purchase cut wood for the stoves due to the wood prices soaring in response to the increased demand for building houses in the post-pandemic global market. Thus, cutting larger pieces of wood into smaller chips using a powered wood splitter (Figure 1) has become more popular among Japanese wood-stove users.

However, incidents and injuries related to wood splitting can occur with the use of these powered tools, such as fractures, amputations, and tissue damage in the upper limbs [3,4]. Hand injuries affect the patient’s social and economic life due to the loss of limb function and the need for long-term rehabilitation. For example, a study from Turkey reported that the commonest cause of machine-related hand injuries was caused by wood splitters [5]. Another database study from the United States, 2.8% of patients hospitalized for traumatic finger amputation from 2012 to 2016 were due to wood splitter [6]. In addition, Lindqvist et al. reported that almost half of patients with wood-splitter-related injuries required finger amputations [7].

Many of the wood splitters used in Japan are imported from overseas, with only a few warnings regarding their use. Powered wood splitters and log-splitting axes are widely available on the Internet and at home-improvement retailers. Although it is critically essential to accumulate clinical information on upper-limb injuries related to wood splitters, the clinical features of these patients remain unclear.

This study aimed to clarify the background and prognosis of such tool-related traumatic injuries, to collect necessary information to discuss preventive measures against hand injuries related to wood splitters.

## 2. Materials and Methods

This was a case-series study including patients with upper-limb injuries related to wood splitting who visited the emergency department at Hirosaki University Hospital from April 2015 to November 2021. Hirosaki University Hospital is located in Tsugaru district, at the northern part of Japan, covering a population of approximately 300,000 people. The emergency department is the only tertiary emergency center in the area; thus, patients with amputated fingers requiring reattachment are transferred to our hospital. In Tsugaru district, apple-farming agriculture is a major primary industry, accounting for about one-fourth of the prefecture’s agricultural products. A large population is engaged in apple-growing agriculture in Tsugaru district. We identified the study patients from our patient databases using the search terms “amputation”, “laceration”, and “fracture.” We identified the patients with wood-splitting-related injuries by excluding those with lower-extremity injuries, trunk injuries, and multiple traumas. Finally, we reviewed these medical records and extracted the data, including the age, gender, occupation, diagnosis, treatment interventions (including surgery), month and time of injury, location, duration of hospitalization, ICU admission, and outcome. In addition, the medical costs for the outpatient and inpatient care for each case were investigated as an indicator of the medical and economic burden. Finally, the patients were interviewed via telephone for information missing from the medical records and the mechanisms of injury. Descriptive statistics summarizing patient characteristics were conducted using Microsoft Excel. These included patient demographics, injury characteristics, medications, emergency department and hospital length of stay, patient charges, and disposition. Data were analyzed at both the patient and incidents levels.

The Graduate School of Medicine Ethics Committee, Hirosaki University, approved this study (ethics approval number 2021-227). Information regarding the study was made available on the hospital’s website to provide information for the participants and the opportunity to opt out from this study.

## 3. Results

We identified 17 cases of upper extremity injuries related to wood splitting among the patients who visited our emergency center during the study period. The results are shown in Table 1. Most of the patients were male (*n* = 15), and their ages ranged from 20 to 81. Most patients were apple farmers (*n* = 6), four were unemployed, and three were office workers. Most of the injuries were to the fingers (*n* = 22), including incomplete amputations (*n* = 5), fingertip amputation (*n* = 5), and laceration (*n* = 7). Other less common but more serious injuries were amputation (*n* = 1) and open fractures (*n* = 4). Injuries to the index finger were the most common (*n* = 9), followed by the thumb in five cases, middle finger in four cases, ring finger in one case, and little finger in three cases (Figure 2). Zone I of the Tamai classification [4] was the most common with five cases, followed by zone IV (*n* = 3), zone II (*n* = 2), and zone III (*n* = 1). Only one case had injury in the proximal area of the elbow joint. The treatments of injury (*n* = 23) included replantation (*n* = 5), amputation (*n* = 5), suture (*n* = 4), open reduction and internal fixation (*n* = 3), composite graft (*n* = 3), tendon suture (*n* = 2), and vascular anastomosis (*n* = 1). Incidents commonly occurred in the daytime of spring (March to May). Eight cases were injured between 9:00 a.m. and 12:00 p.m. and six between 3:00 p.m. and 6:00 p.m. Nine cases received injuries in the spring from March to May, three in the summer, four in the fall, and only one in the winter. Most incidents occurred at home (*n* = 8) or on the patient’s farm (*n* = 6), and no injuries were due to incidents at work.

Six patients required hospitalization, including one patient requiring admission to the ICU. Five patients were referred to other hospitals for further treatments. The median duration of outpatient follow-up was 53 days (IQR 18–194) and 121 days (IQR 53–207) in patients who continued treatment at our hospital. The median length of hospitalization was 18.5 days (IQR 17–21), and the survival rate for replantation (*n* = 5) or vascular repair (*n* = 1) was 83% (*n* = 5). The mean medical cost for replantation was JPY 2,920,680 (Table 2), while for amputation it was JPY 46,433. Fourteen cases were injuries caused by powered wood splitters, two by power saws, and one by a hatchet. The injuries caused by the powered wood splitter were due to the fingers becoming trapped in the machine’s compressor when feeding firewood or supporting firewood with the hand during operation. One patient with an elbow amputation was injured when placing the wood with his elbow. The injury caused by the power saw was due to the firewood popping out while the saw lacked a safety cover. 

## 4. Discussion

This is the first study of wood splitting related upper limb injuries in Japan. This case-series study clarified that powered wood splitters were the leading cause of hand injuries related to wood splitting, and injuries to the index and thumb fingers were common. It was also found that more than half of the patients required significant interventions, such as outpatient follow-up for more than two weeks. 

Although hand injuries are not life-threatening, these injuries could have a significant impact on patients’ well-being due to impaired function and pain in the fingers requiring long-term treatment [8,9]. Our study found that even a laceration requires a long-term outpatient follow-up for as long as seven weeks, and a finger amputation requires more extended outpatient periods, i.e., 7 to 51 weeks. Total hospitalization medical costs for adult severe trauma patients in Japan ranges from JPY1,780,000 to 2,054,000 [10]. We found that injuries due to firewood splitting have medical costs comparable to those of severe trauma. Furthermore, the thumb accounts for 40% of hand function [11,12]. Dias et al. reported that the cost of hand injury is not only the direct medical cost but also the significant impact on the patient’s life, such as lost earnings due to the inability to work and hospital visitation costs [13]. Therefore, it is essential to implement preventive measures for injuries related to wood splitting.

We found that hand injuries related to wood splitting were non-occupational traumas and these results are consistent with previous reports. For example, Lindqvist et al. reported that 7% of the incidents caused by wood splitters occurred during employment, while 85% occurred during the patient’s leisure time [7]. Furthermore, in the United States, power-tool use was a risk factor for hand amputation in non-occupational use among men aged 55 years and older [14]. As in previous reports, private use is suggested to be one of the risk factors for injury.

The three Es (education, enforcement, and improved engineering) policy is a crucial framework for injury prevention [15]. Trybus et al. described that hand injuries could be prevented with proper and consistent safety training, appropriate safety equipment, and the installation of appropriate safety protection [16]. Agencies such as the Ministry of Health and Labor play essential roles in the training and education of employees [17]. On the other hand, personal equipment alone does not provide sufficient safety management and education opportunities. For example, in Japan, the Occupational Safety and Health Regulations require special training for the felling of standing timber, the treatment of dead trees, or lumbering using a chainsaw. However, this does not apply to private use, as the user is responsible for their usage. Another issue is that farmers in Japan are mostly self-employed, and the Occupational Safety and Health Regulations do not apply to them, resulting in a lack of awareness regarding safety management. To prevent trauma, safety training should be provided when these products are sold at retailers, and video training on how to use the products can be provided and made available during Internet sales as well [18].

The mechanism of hand injuries was attributed to the contact with the firewood and the machine itself while the machines are in use. Regarding wood splitters, the wood may jump or become stuck, causing the machine to stop. As specified in the British Standards No. EN 609-1:1999, the instructions state that “the user must not, under any circumstances, touch wood or the machine while it is in operation” [19]. However, the causes of injuries were frequently due to the user’s attempt to support the firewood or to remove the stuck wood while the engine is still running. Therefore, to prevent incidents, we suggest that additional safety measures are essential. For example, attaching covers to the wood splitter to prevent wood from scattering, and adding a function that automatically stops the engine if it becomes stuck. In addition, one patient was injured because the circular saw was an older model and did not have a cover. To prevent such instances of injury, it is also necessary to check the machines regularly and to replace them with new ones. 

There are limitations in this study. First, patients with minor injuries that do not require replantation (e.g., crush) are not included because they are sent to other hospitals. Next, we could not examine the incidence rate of the injuries in our district, the medical or social impact was unknown. Finally, this is a case-series study with small sample size conducted in the single institute; thus, the generalizability of our finding is unclear.

## 5. Conclusions

The wood splitter-related injuries required long-term treatment and often damaged the thumb, a functionally essential digit. All the injuries were sustained during the non-occupational use of a wood splitting. Therefore, safety training should be provided when these products are sold, in order to prevent such traumatic injuries.

## Figures and Tables

**Figure 1 ijerph-19-11507-f001:**
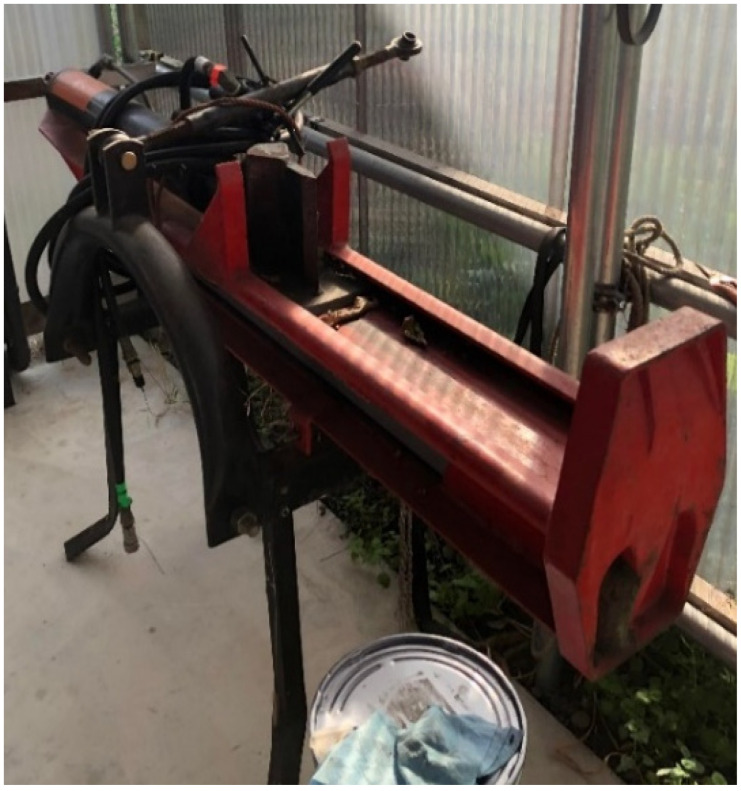
Powered wood splitter.

**Figure 2 ijerph-19-11507-f002:**
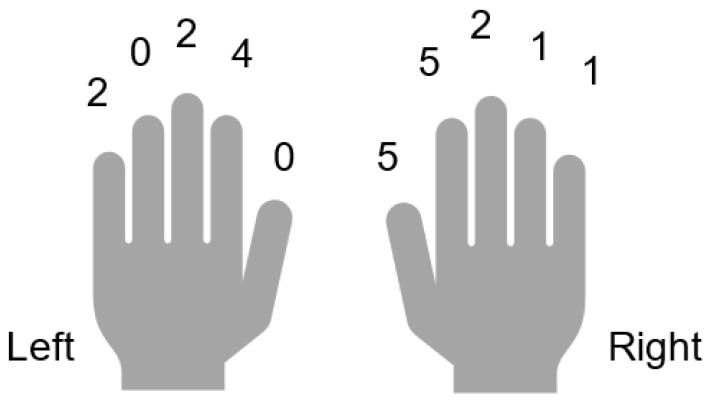
The number of injured fingers.

**Table 1 ijerph-19-11507-t001:** Patient characteristics (17 patients with 22 finger injuries and one elbow amputation).

Variable	
Total cases—no.	17
Male—no. (%)	15 (88.2)
Mean age—years. (Range)	62.4 (20–81)
Occupation—no. (%)	
Office worker	3 (17.6)
Unemployed	4 (23.5)
Farmer	4 (47.1)
Other	2 (11.8)
Injured site—no.	
Finger	22 (95.7)
Other	1 (4.3)
Diagnosis—no. (%)	
Fingertip amputation	5 (21.7)
Finger incomplete amputation	5 (21.7)
Open fracture	4 (17.4)
Laceration	7 (30.4)
Others	2 (8.7)
Tamai classification—no. (%)	
zone I	5 (45.4)
zone II	2 (18.2)
zone III	1 (9.1)
zone IV	3 (27.3)
Mechanism of injury—no. (%)	
Wood splitter	14 (88.2)
Power saw	2 (11.8)
Hatchet	1 (5.9)
Location of incident—no. (%)	
Apple field	6 (35.3)
Yard	2 (11.8)
Home	8 (47.1)
Other	1 (5.9)
Treatment—no. (%)	
Amputation	5 (21.7)
Replantation	5 (21.7)
Composite graft	3 (13.0)
Open Reduction and Internal Fixation	3 (13.0)
Suture	4 (17.4)
Tendon suture	2 (8.7)
Vascular anastomosis	1 (4.3)

M, male: F, female: ORIF, Open reduction, and internal fixation.

**Table 2 ijerph-19-11507-t002:** The mean medical costs and median duration of treatment.

	Replantation	Amputation	Other *	Total
Medical costs (yen)	¥2,920,680	¥46,433	¥262,290	¥993,379
Duration of hospital stay (IQR), days	19 (17–22)	N/A	N/A	18.5 (17–21)
Duration of outpatient care (IQR), days	216 (194–360)	47 (18–53)	51 (16–121)	53 (18–194)

* Other treatments; composite graft, open reduction and internal fixation, suture, tendon suture, vascular anastomosis.

## Data Availability

The data that support the findings of this study are available from the corresponding author upon reasonable request.

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
