# Peer review of "Wood-Splitter-Related Upper-Limb Injuries: A Single-Centered Case-Series Study"

_ijerph, 2022, doi:10.3390/ijerph191811507_

Round 1
Reviewer 1 Report
Abstract.
Please include the time period the data covers in the abstract (ie. April 2015-November 2021).
Introduction. The introduction was informative and the context of increasing use of wood splitters in Japan is of interest to a general readership.
Methods. Some more detail is needed here for an international readership.
Can some brief context be added regarding the likely region covered by the hospital which provided the data. What is size of the population that the hospital services, what are the dominant economic activities in the area that may influence exposure to log splitters (ie. agriculture, horticulture).
I am not sure what the term "we obtained the patients' consent on an opt-out basis" means. Did the authors contact each patient and inform them that they were part of the study, giving them the option to opt out? It isn't entirely clear and needs to be clarified in the manuscript.
This section lacks a description of the analysis undertaken (frequencies and percentages). Please add.
Results.
My main concern with the results is the lack of privacy with regard to publishing the details of the individual cases.
For example Table 1 contains sufficient detail to identify individuals through attribute or identity disclosure. To avoid disclosure of cases the authors need to provide more summative data. Table 1 needs to be removed to protect patient privacy, in addition to specific sentences identifying which cases fit a certain characteristic, such as "Cases 1, 4, 7, 45 8, and 10 were referred to other hospitals for continued treatments." Aggregate cases in Figures 2 & 3 to avoid disclosing cases with only a single count. Table 2 needs the case number column removed.
My other concern is the very basic analysis. The researcher only report frequencies and fail to present percentages. The addition of incident rates would also improve the quality of the analysis.
Discussion.
The discussion lacks a section discussing the strengths and limitations of the research. For example there need to be discussion of the implications of restricting cases with the terms amputations, lacerations & fractures, such as not including crush or other injuries & to upper limb injuries. These limit the generalisability of the findings to other injuries likely to be sustained by wood splitters. Discussion of limitations of small sample size is also needed.
Reviewer 2 Report
Thank you for submitting an interesting article. I have provided some edits and suggestions that will further increase the quality of the manuscript.
- E-mail addresses are not clear in the authors' contact information and line numbering should be consecutive throughout the document.
- Introduction contains previous research and theoretical background but the first paragraph of the introduction should be bibliographically referenced.
- Materials and Methods: The approval number of the ethics committee is missing.
- Results: the figures are very close together. My suggestion is to alternate figures with the text.
- Conclusion: This study aimed to clarify the background, risk factors, and prognosis of trauma to collect the necessary information for discussing preventive measures for hand injuries related to wood splitters, but the conclusions focus on recovery time and location, but not on risk factors or preventive measures to be taken into account.
Reviewer 3 Report
Introduction:
Authors should cite sources for the information in lines 29 - 36.
Consider using more recent citations for finger amputations related to wood splitting: Larsen, M. T., Eldridge-Allegra, I., Wu, J., & Jain, S. A. (2019). Patients admitted for treatment of traumatic finger amputations: characteristics, causes, and prevention. Journal of clinical orthopaedics and trauma, 10(5), 949-953.
Methods:
The authors should state their statistical approach for presenting and analysing the extracted data.
Results: Satisfactory
Discussion: Satisfactory
Round 2
Reviewer 1 Report
Line 79. Replace "encounter" with "incident"
Table 1 - I still have concerns regarding the potential for disclosure of patients in this table of cases included. This type of paper and the resulting table 1 describing a case series is better suited to Trauma Journal, than a public health journal where summative socio-demographic characteristics are expected to decribe cases included.
Line 108. Replace "industrial accidents" with "incidents at work".
Figure 2 & 3. The y-axis label should be "Number of cases". The x-axis is missing a label.
Across whole manuscript - replace the term "accident" with "incident". Other journals ban the use of the work "accident" as it implies that they are unpredictable and unavoidable, rather than predictable and preventable - see https://www.bmj.com/content/322/7298/1320?ijkey=a8777388d9c7a3b0a9ba02e97015ec4c4f03230d&keytype2=tf_ipsecsha
Line 168 - replace "feeling" with "felling".
Author Response
Thank you so much for your careful reviewing.
I. Table
"Table 1 - I still have concerns regarding the potential for disclosure of patients in this table of cases included. This type of paper and the resulting table 1 describing a case series is better suited to Trauma Journal, than a public health journal where summative socio-demographic characteristics are expected to decribe cases included."
In response to your comment, we changed the table1 to the summative one.
"Figure 2 & 3. The y-axis label should be "Number of cases". The x-axis is missing a label."
We decided to remove Figure2 and3 for better information correspondence to new summative table 1. Rathe, we created a new figure of injury sites.
II. Wording comments
"Line 79. Replace "encounter" with "incident""
"Line 108. Replace "industrial accidents" with "incidents at work"."
"Across whole manuscript - replace the term "accident" with "incident". Other journals ban the use of the work "accident" as it implies that they are unpredictable and unavoidable, rather than predictable and preventable -"
Response:
We corrected these points as you suggested.